# Research on Long-Lived Room-Temperature Phosphorescence of Carbazole-Naphthalimide Polylactides

**DOI:** 10.3390/polym12040790

**Published:** 2020-04-02

**Authors:** Zhiwei Li, Xingyuan Zhang

**Affiliations:** CAS Key Laboratory of Soft Matter Chemistry, Department of Polymer Science and Engineering, University of Science and Technology of China, Hefei 230026, China; lvlvlzw@mail.ustc.edu.cn

**Keywords:** polymer long-lived room temperature phosphorescence (RTP), donor-acceptor, intersystem crossing, charge-transfer state, red RTP

## Abstract

Two types of naphthalimide derivatives were synthesized by introducing a carbazole group and an n-butyl, respectively, into the naphthalimide system. The electron-donating ability of two kinds of derivatives was investigated by the electrochemical method. These two types of derivatives were used as initiators for the polymerization of d and l-lactide polymerization. Here, the emission and UV-vis absorption serve as the main focus. Compared with solely donor-initiated polylactide (PLA), the PLA with a donor-acceptor structure has a more efficient phosphorescence emission, of which the longest phosphorescence lifetime is up to 407 ms. The experimental results reveal the existence of charge-transfer states in the donor-acceptor-ended polymer. Due to the role of charge-transfer states, a red phosphorescent polymer was developed. Theoretically, these desirable advantages render synthesized PLAs a potential candidate for bioimaging and anti-counterfeiting.

## 1. Introduction

In recent years, purely organic materials with room temperature phosphorescence (RTP) have attracted increasing attention because of their tremendous applications in chemosensors [1,2], bioimaging [3,4], optoelectronics [5,6,7,8], data storage [9,10], and organic light-emitting diode [11]. A key to realizing efficient RTP is to promote the intersystem crossing (ISC) process. Metal and organic ligand complexes have been a popular design motif because metal to ligand or ligand to metal charge-transfer (CT) states can effectively enhance spin-orbital coupling between well-separated singlet and triplet states. However, these metal–ligand complexes are generally expensive, toxic, and moisture-instable. Therefore, the development of environmentally friendly and easily modified pure organic RTP materials is of urgent importance [12,13]. There have been plenty of reports of pure organic small molecules generating RTP through crystallization [14,15,16,17,18]. For phosphorescent polymer materials, ring-opening polymerization, free radical binary co-polymerization, and covalent cross-linking reactions are considered to be a reasonable chemical strategy to achieve phosphorescent polymer materials. Zhou et al. [19] incorporated a benzophenone derivative with a donor-acceptor structure into an aqueous polyurethane, and the aqueous polyurethane material will exhibit room-temperature phosphorescence. Fraser and Zhang et al. [20] designed a series of dual-emissive polymers with both fluorescence and RTP based on difluoroborn dibenzoylmethane derivatives. Other polymers for organic RTP have been reported [21,22]. These non-doped phosphorescent polymers are derived from the RTP of the polymer itself, with no additional matrix or composite material that facilitates phosphorescence generation. Although some of the above work also produced some pure organic room temperature phosphorescent materials, there are still some problems: The color range of polymer-based RTP is very narrow [23], and there are few reports of red phosphorescent polymer materials, so it is urgent to design and synthesize new phosphorescent units to overcome this problem [24]; compared with inorganic phosphorescent materials, the RTP lifetime of pure organic materials is relatively short (usually <10 ms) [25], so the preparation of long-lived pure organic RTP polymer materials still presents challenges.

Generally, for the naphthalimide structure, the high energy splitting between singlet and triplet states leads to a weak or even unobservable RTP [26]. Inspired by the role of CT states in inducing RTP, a carbazole subunit was introduced into the naphthalimide core to form a donor-acceptor dyad (DAD). By utilizing the CT states as the bridge for the communication between singlet and triplet excitons, the conventional fluorescent naphthalimide is converted into fluorescence and RTP dual emission. Given the conjugation structure of DAD, it is common that aggregation-caused quenching (ACQ) weakens the photoluminescence quantum yield (PLQY) or shuts down the radiative channel. To tackle this notorious issue, the designed DADs are used as initiators for ring-opening polymerization of d, l-lactide, which has more migration-resistant, homogenous luminescence than the traditional physical mixing method. 

Considering that most luminescent materials are used in the solid state, the optical characterization for the synthesized luminescent PLAs will be mainly investigated in the film state. It is found that the photoluminescence of polylactic acid with a donor-acceptor structure is dominated by phosphorescence and that the synthesized carbazole-naphthalimide PLA has a phosphorescence lifetime of up to 407 ms. The synthesis of a red phosphorescent polymer has been achieved. These desirable advantages make synthetic PLA a potential candidate for biological imaging and security.

## 2. Experimental

### 2.1. Reagents and Instruments

The 4-Chloro-1,8-naphthalic anhydride (purity, 94%) was purchased from Energy Chemical Co., Ltd. and used without further purification; 3-amino-9-ethylcarbazole (purity, 95%) was purchased from Macklin Reagent Co., Ltd. and used as received; ethylene glycol (Analytical Reagent )was purchased from Shanghai Titan Science and Technology Co. Ltd.; anhydrous potassium carbonate, dimethyl sulfoxide, 2-mercaptoethanol, n-butylamine, and anhydrous ethanol were purchased from Sinopharm Chemical Reagent Co., Ltd. and used as received. 

All nuclear magnetic resonance (NMR) spectra were recorded on a Bruker AV300 NMR (300 MHz for ^1^H) spectrometer operated in the Fourier transform mode. CDCl_3_ and DMSO-d6 were used as the solvent. NMR chemical shifts were reported in standard format as values in ppm relative to deuterated solvents (CDCl_3_ and DMSO-d6). Molecular weights and molecular weight distributions were determined by gel permeation chromatography (GPC) equipped with Waters 1515 pump and Waters 2414 differential refractive index detector (set at 35 °C). It used a series of two linear Styragel columns (HR2 and HR4) at an oven temperature of 35 °C. The eluent was THF at a flow rate of 1.0 mL/min. A series of low polydispersity polystyrene standards were employed for calibration. Differential Pulse Voltammetry (DPV) experiments were carried out on a electrochemical analyzer (Bioanalysics, CV50W), scan speed: 20 Mv/s. UV-vis-NIR absorption spectra were recorded on a SOLID3700 UV-vis-NIR spectrometer. Steady-state emission and delayed photoluminescence spectra were conducted on a F-4600 (Hitachi) spectrofluorometer, the slit widths were both set at 5 nm for excitation and emission. Fluorescence and phosphorescence lifetime data were acquired with a 1 MHz LED laser with the excitation peak at 370 nm. Lifetime data were analyzed with DataStation v6.6 (Horiba Scientific).

### 2.2. Synthesis

#### 2.2.1. Synthesis of 6-chloro-2-(9-ethyl-9H-carbazol-3-yl)-1H-benzo[de]isoquinoline-1,3(2H)-dione(A)

The 4-Chloro-1,8-naphthalic anhydride (1 g) and 3-amino-9-ethylcarbazole (1.367 g) (molar ratio of 1: 1.5) were added to a 250 mL single-necked flask, and 50 mL of ethylene glycol was used as a solvent. The reaction was carried out at a temperature of 120 °C overnight, and the reaction was monitored by TLC technology. After the reaction was completed, it was cooled to room temperature, and sufficient methanol was added. When precipitation occurred, suction filtration and drying was used to obtain product A (Scheme 1) (1.45 g), with a yield of 79.4%. ^1^H NMR (300 MHz, DMSO-d6) δ = 8.69 (d, J = 8.5, 1H), 8.64 (d, J = 7.2, 1H), 8.48 (d, J = 7.9, 1H), 8.17 (d, J = 1.9, 1H), 8.09 (dt, J = 16.9, 6.9, 3H), 7.74 (d, J = 8.6, 1H), 7.68 (d, J = 8.3, 1H), 7.53–7.44 (m, 2H), 7.22 (t, J = 7.4, 1H), 4.53 (q, J = 7.0, 2H), 1.38 (t, J = 7.1, 3H).

#### 2.2.2. Synthesis of 2-butyl-6-chloro-1H-benzo[de]isoquinoline-1,3(2H)-dione(B)

A total of 1 g of 4-chloro-1,8-naphthalic anhydride was put into a 250 mL single-necked flask, and then 50 mL of absolute ethanol was added. Then, 0.6278 g of n-butylamine was diluted in 5 mL of absolute ethanol and was added to the reaction system. The reaction time ended after 12 h. After the reaction was cooled to room temperature, a large amount of deionized water was added. When a solid precipitated out, it underwent suction filtration, washing with deionized water and absolute ethanol, and drying to obtain product B (Scheme 1) (0.545 g) with a yield of 44%. ^1^H NMR (300 MHz, CDCl3) δ = 8.65 (t, J = 9.1, 1H), 8.59 (d, J = 8.5, 1H), 8.53–8.46 (m, 1H), 7.91–7.77 (m, 2H), 4.21–4.14 (m, 2H), 1.78–1.66 (m, 2H), 1.52–1.38 (m, 2H), 0.98 (t, J = 7.3, 3H).

#### 2.2.3. Synthesis of 2-(9-ethyl-9H-carbazol-3-yl)-6-(2-hydroxyethoxy)-1H-benzo[de]isoquinoline-1,3(2H)-dione (CZ-NI-OH)

A total of 1 g of A and 1.3248 g of anhydrous potassium carbonate (molar ratio of 1:4) were added to a 250 mL single-necked flask, and 50 mL of ethylene glycol was added; this was reacted at a temperature of 120 °C for 12 h and was monitored by TLC reaction. When the reaction was complete, the entire reaction system was cooled to room temperature and a large amount of deionized water was added. The mixture was stirred at room temperature with a stirrer until the solid precipitated, and this was then filtered with suction. The obtained solid was placed in a vacuum oven to obtain the substance CZ–NI–OH (Scheme 1) (0.72 g), with a yield of 66.6%. ^1^H NMR (300 MHz, DMSO-d6) δ = 8.72 (dt, J = 17.1, 8.6, 1H), 8.59–8.43 (m, 2H), 8.19–8.08 (m, 2H), 7.88 (dd, J = 8.3, 7.4, 1H), 7.76–7.63 (m, 2H), 7.55–7.32 (m, 3H), 7.21 (t, J = 7.4, 1H), 5.16 (t, J = 5.8, 1H), 4.60–4.45 (m, 2H), 4.38 (t, J = 4.6, 2H), 3.94 (dd, J = 9.7, 5.3, 2H), 1.38 (t, J = 7.1, 3H).

#### 2.2.4. Synthesis of 2-(9-ethyl-9H-carbazol-3-yl)-6-((2-hydroxyethyl)thio)-1H-benzo[de]isoquinoline-1,3(2H)-dione (CZ-NI-SH)

A total of 1 g of intermediate A, 0.28 g of 2-mercaptoethanol, and 1.3248 g of anhydrous potassium carbonate (molar ratio 1:1.5:4) were placed in a 250 mL single-necked flask, and 50 mL of dimethyl sulfoxide (DMSO) was added. The reaction temperature was 50 °C and the reaction time was 3 h. After the reaction was complete, as in the previous treatment, a sufficient amount of deionized water was added, and this was stirred until a solid precipitated (if there was no solid precipitate, the extraction was performed with dichloromethane), which was then filtered with suction. The precipitated solid was dried in a vacuum oven to obtain the product CZ-NI-SH (Scheme 1) (0.695 g), with a yield of 62.1%. ^1^H NMR (300 MHz, DMSO-d6) δ = 8.63 (d, J = 8.5, 1H), 8.56 (d, J = 7.2, 1H), 8.41 (d, J = 7.9, 1H), 8.18–8.09 (m, 2H), 7.93 (t, J = 7.9, 1H), 7.84 (d, J = 8.0, 1H), 7.69 (dd, J = 21.9, 8.5, 2H), 7.53–7.39 (m, 2H), 7.21 (t, J = 7.4, 1H), 5.18 (t, J = 5.5, 1H), 4.52 (q, J = 6.9, 2H), 3.78 (q, J = 6.1, 2H), 3.39 (dd, J = 12.8, 6.4, 2H), 1.38 (t, J = 7.0, 3H).

#### 2.2.5. Synthesis of 2-butyl-6-(2-hydroxyethoxy)-1H-benzo[de]isoquinoline-1,3(2H)-dione(NI-OH)

A total of 1 g of intermediate B, 1.932 g of anhydrous potassium carbonate (molar ratio 1:4) and 50 mL of ethylene glycol were added to a 250 mL single-necked flask; this was reacted at a temperature of 130 °C for 12 h and the reaction was monitored by TLC. Finally, the system was cooled to room temperature, and then a large amount of deionized water was added and stirred until a solid precipitated out. After suction filtration was complete, the product was dried under a vacuum to obtain the product NI-OH (Scheme 1) (0.487 g), with a yield of 44.4%. ^1^H NMR (300 MHz, DMSO-d6) δ = 8.63 (d, J = 8.4, 1H), 8.45 (dd, J = 25.4, 7.8, 2H), 7.81 (t, J = 7.8, 1H), 7.29 (d, J = 8.3, 1H), 5.11 (t, J = 5.7, 1H), 4.33 (t, J = 4.6, 2H), 4.08–3.96 (m, 2H), 3.91 (dd, J = 9.8, 5.0, 2H), 1.68–1.54 (m, 2H), 1.43–1.30 (m, 2H), 0.93 (t, J = 7.3, 3H).

#### 2.2.6. Synthesis of 2-butyl-6-((2-hydroxyethyl)thio)-1H-benzo[de]isoquinoline-1,3(2H)-dione(NI-SH)

A total of 1 g of Intermediate B, 0.4095 g of 2-mercaptoethanol, 1.932 g of anhydrous potassium carbonate (molar ratio of 1:1.5:4), and 50 mL of dimethyl sulfoxide as solvent were reacted at a temperature of 50 °C; the reaction time was 3 h, the same as for the previous step. After the reaction was complete, the reaction was withdrawn, and it was cooled to room temperature. We added a large amount of deionized water and then stirred until a solid precipitate appeared in the system, and the precipitated solid was obtained by suction filtration. The product NI-SH (Scheme 1) was obtained by drying (0.523 g), with a yield of 45.4%. ^1^H NMR (300 MHz, DMSO-d6) δ = 8.58–8.50 (m, 2H), 8.36 (d, J = 7.9, 1H), 7.88 (dd, J = 8.4, 7.4, 1H), 7.79 (d, J = 8.0, 1H), 5.16 (t, J = 5.5, 1H), 4.07–3.99 (m, 2H), 3.75 (q, J = 6.2, 2H), 3.37 (d, J = 6.5, 2H), 1.67–1.55 (m, 2H), 1.34 (m, 2H), 0.93 (t, J = 7.4, 3H).

#### 2.2.7. Synthesis of PLAs

To obtain CZ-NI-OH: D, L-lactide, stannous octoate was used at a molar ratio of 1:150:0.1. We took 0.5 g of d, l-Lactide, put it into a 10 mL Knotes bottle, passed it through nitrogen, made the whole reaction system oxygen-free, then added stannous octoate diluted with n-hexane and immersed the whole ball of the flask. A viscous solid formed in an oil bath at 135 °C. After the mixture was cooled to room temperature, the crude polymer was purified by precipitation from CH_2_Cl_2_/ice CH_3_OH (×3) at this temperature. The resulting solid was further precipitated from CH_2_Cl_2_/hexane (×3) to give a final polymer [27], CZ-NI-O-PLA (Scheme 2). Following the same operation, three other polylactides were sequentially synthesized: CZ-NI-S-PLA, NI-O-PLA, and NI-S-PLA (Scheme 2).

## 3. Result and Discussion

### 3.1. Characterization of Electrochemical Properties of Four Derivatives

According to the synthesis steps mentioned, we synthesized four kinds of monohydroxy monomers (Scheme 1), and the ^1^H NMR results verified the correctness of their structures.

We research the electrochemical properties of small molecules and use Differential Pulse Voltammetry (DPV) to test these four small molecules. The curve of DPV is more refined than the conventional Cyclic Voltammetry (CV).

The DPV curves of these four molecules (Figure 1) have peaks at positive potentials. The oxidation potentials of the molecules NI-OH and NI-SH are around 1.6 eV, while the oxidation potentials of CZ-NI-OH and CZ-NI-SH are about 1.2 eV. Combining these four molecular structures, the electron-deficient sites of the molecules are all naphthalimide structures, so the reduction potentials of all molecules are around −1.6 eV. Compared with molecules CZ-NI-OH and NI-OH, the oxidation potential of CZ-NI-OH is lower and the donor ability is stronger. Through electrochemical experiments, the experimental results are consistent with our previous conjectures. When a molecule has a donor-acceptor structure, the lower the oxidation potential, the stronger the electron donor capacity, and the easier the charge separation is to produce an intramolecular charge transfer state. 

### 3.2. Photoluminescence Properties of Polylactides

Under a nitrogen atmosphere, these four functional monomers were subjected to ring-opening polymerization (ROP) with D, L-lactide at 130 °C to obtain polylactides. The Mn of PLA was obtained from the GPC spectrum (Table 1). The data show that the PDI of CZ-NI-O-PLA and CZ-NI-S-PLA is slightly larger than the other two types of polylactic acid. These results are attributed to the tendency of naphthalimide to form ground state dimers, which may hinder the ROP process and cause a wider distribution of *M*_w_ [27]. 

Considering that phosphorescence will be quenched by oxygen or under solution conditions, the four kinds of polylactides mentioned above were dissolved in CH_2_Cl_2_; then the solution mixture was dried in a glass tube to form a transparent uniform film (ca. 0.2 mm).

NI-O-PLA emits dazzling blue fluorescence under the illumination of a handheld 365 nm UV lamp. Regardless of the steady-state emission spectrum under air or vacuum conditions (Figure 2a), the fluorescence emission spectrum of the polylactide shows almost no change. The fluorescence lifetime of the NI-O-PLA emission wavelength at 420 nm is 7.14 ns (Figure 3a, air, 298 K). The two characteristics of the steady-state emission spectrum and luminescence lifetime data show that the photoluminescence of NI-O-PLA is mainly fluorescence emission. To attempt to enhance the phosphorescence emission of this system, a strong electron-donating group carbazole was introduced into the naphthalimide system to obtain a donor-acceptor covalently-linked polylactide (CZ-NI-O-PLA). Figure 2b is its steady-state emission spectrum. Obviously, the steady-state emission spectrum shows completely different states under air and vacuum conditions. The emission of this polylactide film in the air is mainly fluorescent, the maximum emission wavelength is 442 nm, and the lifetime at this emission wavelength is 5.381 ns (Figure 3b, air, 298 K). When the glass tube was sealed in a vacuum, the maximum emission wavelength shifts from 442 to 550 nm, and the luminescence lifetime at this wavelength is 407 ms (Figure 3c, vacuum, 298 K). The vacuum film of CZ-NI-O-PLA emits a bright yellow phosphorescence under the irradiation of a handheld 365 nm UV lamp. After removing the excitation light source, the film will have a visible delay, which also explains why this polylactide film has a longer phosphorescent lifetime. The difference between the photoluminescence spectrum of CZ-NI-O-PLA under air and vacuum conditions and the luminous lifetime are all characteristics of phosphorescence emission. In its RTP spectrum (Figure 2b, blue curve), the fluorescence emission peak disappears, and only 550 and 595 nm phosphorescence peaks show up, which further proves that the emission of CZ-NI-O-PLA at 550 and 595 nm is phosphorescence emission. The RTP spectrum shows the vibration of the aromatic ring [26], that is, the emission peaks of 550 and 595 nm are separated by about 1500 cm^−1^. In the conventional concept, phosphorescence is quenched under air conditions. Generally, there is no phosphorescent peak in the steady-state spectrum under air conditions, but here we see that at the same location, there is still a phosphorescent emission peak in the steady-state emission spectrum in the air. The reason for this phenomenon is that during the preparation of the polylactide film, the formed film is dense enough. At the same time, the emission of phosphorescence is also strong, and the oxygen content in the film is not sufficient to quench the phosphorescence. When the excitation source is continuously irradiated, a similar phenomenon occurs in the film. We irradiated the CZ-NI-O-PLA film in air by a 365 nm hand-held UV lamp, with the strong yellow luminescence under continuous irradiation. 

Since the CZ-NI-O-PLA phosphorescent color is yellow, it is easy to achieve red phosphorescence. Considering the close nature of the O atom and the S atom, we replaced the O atom of the above molecule with the S atom to obtain NI-S-PLA and CZ-NI-S-PLA. The photoluminescence of the NI-S-PLA film (Figure 2c) is basically the same as that of NI-O-PLA. The maximum emission wavelength under air and vacuum is 445 nm. The luminescence lifetime at this wavelength is 5.91 ns (Figure 3d, air, 298 K). Substance CZ-NI-S-PLA emits bright red phosphorescence under the excitation light source of 365 nm. The lifetime decay spectrum (Figure 3e,f) shows that the emission of this substance at 450 nm under air conditions is fluorescent emission, while the emission peaks at 590 and 635 nm in vacuum are phosphorescent peaks (Figure 2d). Comparing Figure 2b,d, the shapes of the curves are similar. However, the maximum emission wavelength of CZ-NI-S-PLA is slightly longer. Comparing the structures of CZ-NI-O-PLA and CZ-NI-S-PLA, the difference between these two polylactides structures is the O atom and the S atom. As the lone pair of electrons of the S atom forms a p-π conjugate with naphthalimide and the electrons of the entire system are delocalized to a greater extent, it is more likely to obtain red phosphorescence with lower energy, and the maximum emission wavelength of CZ-NI-S-PLA is longer. This provides a way to realize the red phosphorescent unit. Compared with the previously reported red phosphorescent unit, this unit has better solubility, is simpler and synthesis is more efficient. As red phosphorescence has a stronger penetrating ability and PLA has good biocompatibility, it has good application prospects in the field of cell imaging. Comparing the photoluminescence of these four covalently linked polylactides, we can conclude that the presence or absence of the donor-acceptor structure will have an important effect on the phosphorescence emission. Among them, NI-O-PLA and NI-S-PLA are mainly fluorescence emission, while CZ-NI-O-PLA and CZ-NI-S-PLA are mainly phosphorescence emission. Considering the difference in structure, NI-O-PLA is an electron-deficient system. However, due to the effect of carbazole, CZ-NI-O-PLA is prone to generate intramolecular charge transfer states to bridge the splitting energy between larger singlet state and triplet state (Δ*E*_ST_), thus promoting the intersystem crossing process. Therefore, in the photoluminescence process of CZ-NI-O-PLA, phosphorescent emission is in a dominant position to compete with fluorescence emission, resulting in the photoluminescence of the polylactide film being dominated by phosphorescence emission.

The photoluminescence properties of PLA films have been discussed. As PLA is soluble in most organic solvents, these polymeric conjugates also exhibit excellent solubility in polar solvents such as tetrahydrofuran (THF) and dichloromethane (DCM). We choose four different solvents (DCM, EA, THF, DMF). The corresponding steady-state emission spectra for the two phosphorescent polymers (CZ-NI-O-PLA, CZ-NI-S-PLA) were recorded in different solvents, which influence the luminescence properties. As can be seen, the emission maximum varies from 432 to 438 nm for CZ-NI-O-PLA (Figure 4a) and 432 to 439 nm for CZ-NI-S-PLA (Figure 4b). The solvatochromic fluorescence is typically a sign of charge transfer state [27], which is expected for the donor-acceptor type of molecules. Although the maximum emission peaks of these two phosphorescent PLA have shifted, the phenomenon of this shift is not obvious. The possible reason is that under such conditions, the CT state is a dark state, and the test under solution conditions will not have a significant shift. Therefore, we should provide more evidence to prove the CT state.

### 3.3. UV Absorption Properties of Polylactic Acid and Verification of Intramolecular Charge Transfer State

Four kinds of polylactic acid are dissolved in dichloromethane to control the concentration of chromophore to 10^−5^ mol/L. Observing the ultraviolet spectrum (Figure 5a), compared with NI-O-PLA, the UV spectrum of CZ-NI-O-PLA has a small absorption peak at 290 nm, which is attributed to the contribution of the carbazole structure. Methods for proving charge transfer state have been proposed in a large number of previous studies [28,29]. Among them, the most common and simple method is to identify it by the emission and absorption of different polar solvents [30]. We also tried to use this method to research CZ-NI-O-PLA and CZ-NI-S-PLA. Polylactide was dissolved in solvents of different polarities (order of polarity of the solvent: DCM < THF < EA < DMF), and the concentration of chromophore was controlled at 10^−5^ mol/L. Figure 5b,c shows the UV-vis spectra of CZ-NI-O-PLA and CZ-NI-S-PLA under different polar solvents. It was found that the absorption peak of CZ-NI-O-PLA will redshift as the polarity of the solvent increases (emission wavelength: λ_DMF_ > λ_EA_ > λ_THF_ > λ_DCM_).

In the transition band of π–π* in the ultraviolet spectrum, the absorption peak is red-shifted from 235 to 270 nm. CZ-NI-S-PLA will show a similar phenomenon in the same operation. This phenomenon proves that CZ-NI-O-PLA and CZ-NI-S-PLA have the characteristics of intramolecular charge transfer state, and the areas and characteristics of redshift are significant features for π–π* transitions.

Taking the UV absorption in different polar solvents into account, the experimental results appear to confirm our conjecture. Then, the emission of these two polylactides in different polar solvents should also have this phenomenon, but the shift of the emission peak (Figure 4a,b) is not as obvious as we thought. The possible reason is that under such conditions, the CT state is a dark state, and the test under solution conditions will not have a significant shift. Considering the existence of this problem, we changed our thinking to verify the emission of CZ-NI-O-PLA and CZ-NI-S-PLA in different polar solvents.

Chromophores are dispersed in different polymer matrices. Polymethyl methacrylate (PMMA), polystyrene (PS), poly N, N-methyl dimethacrylate (PDMA), and polyurethane (PU) were selected as different polar dispersive matrices. We dispersed the small molecules CZ-NI-OH and CZ-NI-SH in the above four polymer matrices, with a mass fraction of 1%. Then, 10 mL of solvent DCM was used to dissolve it, and this was then blown into a glass tube to form a transparent and uniform film. It is believed that the chromophore is dissolved in a polymer matrix of different polarities. The fluorescence emission spectrum of small molecules dispersed in four polymer matrices (Figure 6a,b) is the main evidence to prove the CT state. The steady-state emission peaks of the blend in air increased sequentially from PMMA to PU, and the positions of the emission peaks shifted to red as the polarity of the polymer matrix increased (λ_PU_ > λ_PS_ > λ_PDMA_ > λ_PMMA_). In this operation, considering that when small molecules are blended into the polymer, the entire system is not stable, and small molecules will have different interactions (such as π–π, H-bonds, etc…) with the polymer matrix. For this reason, we use the fastest pace possible when testing. Although we have performed this process as quickly as possible, in fact, we could not control the interaction between the matrix and the dye; therefore, we chose different polymers as dispersed matrices in order to qualitatively compare the influence of the matrices. In combination with the absorption and emission spectra in different polar solvents, the emission in different polymer matrices can be used as an auxiliary to prove the existence of charge transfer states.

Based on the above test results of various forms of ultraviolet absorption and fluorescence emission, we firmly believe that the small molecules CZ-NI-OH and CZ-NI-SH exist in the intramolecular charge transfer state.

## 4. Conclusions

In this work, the synthesized polylactide with carbazole-naphthalimide structure has the characteristics of long-lived room temperature phosphorescence emission, and its lifetime can be up to 407 ms; by introducing different groups to the naphthalimide system, the fluorescence/phosphorescence emission ratio of this type of polylactide can be tuned. Among the two types of polylactic acid mainly dominated by phosphorescent emission, O and S atoms will also have an important effect on the system. Polylactide containing O atoms has a longer phosphorescence lifetime, while the maximum emission wavelength of polylactide containing S atoms is relatively longer, and it is more likely to produce red room-temperature phosphorescence.

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
