# Peer review of "Research on Long-Lived Room-Temperature Phosphorescence of Carbazole-Naphthalimide Polylactides"

_polymers, 2020, doi:10.3390/polym12040790_

Round 1

Reviewer 1 Report

The manuscript entitled “Research on Long-lived Room-Temperature Phosphorescence of Carbazole-Naphthalimide Polylactides” by Zhiwei, Li and Xingyuan, Zhang is interesting and deserves publication.

The authors were able to synthesize 4 Polylactides using 4 different ROP initiators. Two of these polymers are phosphorescent at room temperature. Those phosphorescent polymers bear at the chain ends

The phosphorescent polymers bear a carbazole- naphthalimide dyad at one chain end that is responsible for the emission. The phosphorescence arises from the enhancement of the ISC due to second order spin-orbit charge transfer mechanism.

The authors should complement the results by including:

  1. the quantum yields of Fluorescence, Phosphoresce and ISC of the polymers.
  2. 2) The authors should clarify what they men for “room temperature”. What is the difference between vacuum and Room temperature (RT)?
  • the authors should explain the difference between spectra in both Fig 2 b Fig 2d at room temperature and in Vacuum.
  1. the authors should clarify along text and in Fig.3 caption the conditions at which were recorded the decays.
  2. An explanation should be done for the difference in lifetimes in Fig.3 c) and Fig3 f) since the two wavelengths correspond to 2 vibronic peaks of the same electronic transition.
  3. The solvatochromic shifts analyses should appear after the discussion of the spectra and not at the end of the manuscript.
  • The difference in the shift in the absorption and luminescence spectra are generally interpreted in terms of the difference in dipole moment in the ground and excited state. The interpretation given by the authors does not seems adequate.
  • The shifts in different polymer matrices are not appropriate to discuss the nature of charge transfer states because the interaction with the matrix can be different . This is the case (Fig. 5c) and d) ) since the shape of the spectra differs with the matrix and so the results are not conclusive.

In conclusion, the manuscript can be reconsidered after major revision and after a full revising of the scientific writing.

Author Response

Dear. reviewer:

Thank you for reviewing our manuscript in your busy schedule. Your suggestions are very constructive.Our point-by-point responses are in the attachment below.

Best wishes.

Zhiwei Li. 

Reviewer 2 Report

In this work, the authors show how it is possible to obtain phosphorescent emission by incorporating strong electron-donating group carbazole to the molecular structure. This process is known and we can find research in the same line in the scientific literature. Despite this, the referee considers that the data set out in this work are interesting in the sense of the synthesis of phosphorescent polymers stable at room temperature. Once the corrections are made, the work can be published.

  1. Line 15. PLA is not defined.
  2. Line 18. “de-sirable” must be changed by desirable.
  3. Line 19. “an-ti-counterfeiting” must be changed by anti-counterfeiting.
  4. Line 67. There is space between 1.8- and naphthalic in “4-Chloro-1,8- naphthalic anhydride”. It must be rewritten as 4-Chloro-1,8-naphthalic anhydride.
  5. Line 167. “while the oxidation of CZ-NI-OH and CZ-NI-SH The potential is around 1.2 eV”. the phrase must be rewritten correctly.
  6. Lines 190-220. The emission spectra are shown in the Figure 2 in air, vacuum and room temperature conditions. Please specify that they are air conditions. When the spectra are obtained in air, what is the temperature?.
  7. Figure 2. The PL emission in vacuum for CZ-NI-S-PLA derivate is somewhat larger than CZ-NI-O-PLA derivative. What is the reason?.
  8. Lines 235 – 236. Why is the synthesis of derivatives with S easier?.
  9. Lines 255 – 256. The macroscopic parameter that shows the polarity of a solvent is the dielectric constant (ε). Please. indicate the values ​​of ε for each solvent and order in increasing polarity value.
  10. Line 257. “Figure 4 (b c)”. It must changed by Figure 4b and 4c.
  11. Line 271. The authors should define that it is a dark state since they conjecture that the non-existence of shift in the PL emission is due to the existence of states of this type.
  12. Line 274. Should they be different polymeric matrices with different polarity?.
  13. Lines 278 – 280. Please. indicate the values ​​of ε for each polymer matrix.
  14. Line 284. “(Figure 5 c d)”. It must changed by Figure 5c and 4d.
  15. Lines 278 – 292. When small fluorescence molecules are dispersed in a polymeric matrix, different types of interaction can be occurs (π – π, H – bonds, etc…). The authors not comment anything about this. The sentence should be improved.

Author Response

Dear reviewer:

Thanks for reviewing our manuscript in your busy schedule.Your comments and advices are very constructive, and we appreciate your attention on details. Our point-by-point responses are in the attachment below.

Best wishes 

Zhiwei Li.

Round 2

Reviewer 1 Report

The authors should revise the scientific English writing. Following my previous comments on the solvatochromic shifts the authors must improve the discussion.

Author Response

Dear reviewer:

We greatly appreciate your valuable suggestions for our work. Your opinion has a great effect on the logic of the entire article.Our point-by-point response is given below.

Comment: The authors should revise the scientific English writing.Following my previous comments on the solvatochromic shifts the authors must improve the discussion.

Our reply: thanks for pointing out this point.We have revise some English writing. And based on your previous comments on solvatochromic shifts,we re-arrange the structure of paper.The solvatochromic shifts analyses appear after the discussion of the spectra,and we improve some discussion about solvatochromic. Line 264-275.

Finally,we would like to express our heartfelt thanks to the reviewer for your suggestions. It is your suggestions that make this article better.

Best wishes 

Zhiwei Li.